# Co-Stimulation of Purinergic P2X4 and Prostanoid EP3 Receptors Triggers Synergistic Degranulation in Murine Mast Cells

**DOI:** 10.3390/ijms20205157

**Published:** 2019-10-17

**Authors:** Kazuki Yoshida, Makoto Tajima, Tomoki Nagano, Kosuke Obayashi, Masaaki Ito, Kimiko Yamamoto, Isao Matsuoka

**Affiliations:** 1Laboratory of Pharmacology, Faculty of Pharmacy, Takasaki University of Health and Welfare, Takasaki-shi, Gunma 370-0033, Japan; yoshida-k@takasaki-u.ac.jp (K.Y.); 1321057@takasaki-u.ac.jp (M.T.); 1321067@takasaki-u.ac.jp (T.N.); 0821024@takasaki-u.ac.jp (K.O.); mito@takasaki-u.ac.jp (M.I.); 2Department of Biomedical Engineering, Graduate School of Medicine, The University of Tokyo, Tokyo 113-0033, Japan; k-yamamoto@umin.ac.jp

**Keywords:** extracellular ATP, P2X4 receptor, prostaglandin E_2_, EP3 receptor, bone marrow-derived mast cell, mast cell degranulation, Ca^2+^ influx, PI3 kinase

## Abstract

Mast cells (MCs) recognize antigens (Ag) via IgE-bound high affinity IgE receptors (FcεRI) and trigger type I allergic reactions. FcεRI-mediated MC activation is regulated by various G protein-coupled receptor (GPCR) agonists. We recently reported that ionotropic P2X4 receptor (P2X4R) stimulation enhanced FcεRI-mediated degranulation. Since MCs are involved in Ag-independent hypersensitivity, we investigated whether co-stimulation with ATP and GPCR agonists in the absence of Ag affects MC degranulation. Prostaglandin E_2_ (PGE_2_) induced synergistic degranulation when bone marrow-derived MCs (BMMCs) were co-stimulated with ATP, while pharmacological analyses revealed that the effects of PGE_2_ and ATP were mediated by EP3 and P2X4R, respectively. Consistently, this response was absent in BMMCs prepared from P2X4R-deficient mice. The effects of ATP and PGE_2_ were reduced by PI3 kinase inhibitors but were insensitive to tyrosine kinase inhibitors which suppressed the enhanced degranulation induced by Ag and ATP. MC-dependent PGE_2_-triggered vascular hyperpermeability was abrogated in a P2X4R-deficient mouse ear edema model. Collectively, our results suggest that P2X4R signaling enhances EP3R-mediated MC activation via a different mechanism to that involved in enhancing Ag-induced responses. Moreover, the cooperative effects of the common inflammatory mediators ATP and PGE_2_ on MCs may be involved in Ag-independent hypersensitivity *in vivo*.

## 1. Introduction

Mast cells (MC) are widely distributed in the body and abundant in tissues that contact with the external environment, such as the intestine, respiratory tract, and skin [1]. MCs are replete with secretory granules containing a variety of preformed mediators, such as histamine, cytokines, and proteases, that they release in response to various harmful pathogens or invading external substances [2]. This process is thought to initiate immunoregulatory reactions by providing a microenvironment that recruits and activates other immunocompetent cells; however, such reactions are sometimes inappropriately enhanced under certain conditions, causing hypersensitivity and allergic inflammation [3]. The most well-known MC activation pathway involves high affinity IgE receptors (FcεRI), which enable MCs to form a barrier against pathogen invasion [4]. When antigens (Ag) cross-link the IgE-FcεRI complex, many signaling molecules are phosphorylated by the protein tyrosine kinase Lyn and subsequently recruited Syk, activating multiple pathways involving processes such as degranulation, cytokine production, and lipid mediator production [5]. MCs are known to express various immunoregulatory receptors, including toll-like receptors, stem cell factor (SCF) receptors, and G-protein-coupled receptors (GPCR), whose stimulation has been reported to promote IgE-dependent MC degranulation [6]. For instance, it has been shown that PGE_2_, which is known to mediate inflammation [7], and adenosine, which accumulates extracellularly under ischemic conditions [8], enhance Ag-mediated MC degranulation [9,10,11]. The effects of PGE_2_ and adenosine are mediated by the EP3 receptor (EP3R) and A_3_ receptor (A_3_R), respectively, and are commonly transmitted via the pertussis toxin (PTX)-sensitive Gi protein, suggesting a similar underlying mechanism. Indeed, several studies have demonstrated that enhancing Ag-induced MC degranulation by co-stimulation with Gi-coupled receptor agonists requires the activation of phosphoinositide 3-kinase (PI3K) γ, a PI3K subclass regulated by the Gi protein βγ subunit [10,11]. These enhanced responses are thought to be involved in exacerbating allergic reactions.

In addition to Gi-coupled receptor agonists, we recently reported that extracellular ATP also enhances the Ag-induced degranulation response in bone marrow-derived MCs (BMMCs) by activating the P2X4 receptor (P2X4R), a ligand-gated ion channel [12]. Unlike Gi-coupled receptor agonists, P2X4R stimulation does not induce the PI3K signaling pathway, but enhances the Ag-induced tyrosine phosphorylation of signaling molecules including Syk and phospholipase C (PLC) γ [13]. Based on these results, we hypothesized that stimulation of MCs with ATP and Gi-coupled receptor agonists may cause MC degranulation in an IgE-independent manner. Indeed, we previously showed that co-stimulating BMMCs with ATP and adenosine induced synergistic degranulation via A_3_R [12].

With respect to IgE-independent MC activation, MCs express several pattern recognition-receptors, such as toll-like receptor 2 and 4, which are stimulated by micro-organism specific molecular motif, triggering the innate immune responses [14]. In this category, extracellular ATP is also considered to act as a danger signal, because living cells contain high concentrations of ATP and release it to extracellular space under adverse conditions, like tissue damage, necrosis, and pyroptosis [15]. The accumulation of ATP is detected by surrounding cells through a wide variety of receptors; not only G protein-coupled receptors (P2Y_1,2,4,6,11,12,13,14_), but also ionotropic receptors (P2X1-7). Upon stimulation through different mechanism, MC is known to be capable of producing a wide variety of inflammatory and anti-inflammatory cytokines, no less than other immune cells such as macrophages and lymphocytes [16]. Indeed, MCs have quite recently been implicated in the pathogenesis of systemic lupus erythematosus [17] and neurofibromatosis [18]. In such inflammatory diseases, MC activation cannot be explained only by the IgE-dependent mechanism. Since, extracellular ATP is suggested to be released from cells via mechanical stress such as itching-induced scratching behavior [19,20], MCs must be exposed to ATP and various humoral factors in the inflammatory environment.

This study therefore investigated whether combining ATP with GPCR agonists affected MC degranulation, finding that co-stimulation with ATP and PGE_2_ induced synergistic MC degranulation by activating P2X4 and EP3R, respectively, via a novel mechanism that is different to the Ag-induced response.

## 2. Results

### 2.1. Effects of ATP and GPCR Agonist co-Stimulation on BMMC Degranulation

We first examined the effect of ATP and GPCR agonist co-stimulation on the degranulation of BMMCs. We tested the GPCR agonists sphingosine-1-phosphate (S1P) (1 μM), PGE_2_ (1 μM), histamine (100 μM), C5a (10 nM), PGD_2_ (1 μM), UDP-glucose (100 μM), and compound 48/80 (10 μM), all of which constantly increased the intracellular Ca^2+^ concentration ([Ca^2+^]i) of Fura-2 loaded BMMCs, albeit to varying degrees (data not shown). Although these GPCR agonists had no effect on BMMC degranulation when tested alone, PGE_2_ markedly induced degranulation when concurrently stimulated with 100 μM ATP (Figure 1A). Time course experiments revealed that co-stimulation with ATP and PGE_2_ induced degranulation rapidly, with the response initiated within 2 min and reaching the maximum steady state in 5 min (Figure 1B). Different PGE_2_ concentrations only induced BMMC degranulation in a concentration-dependent manner in the presence of 100 μM ATP, with PGE_1_ inducing similar effects (Figure 1C). Moreover, the effects of ATP on degranulation in the presence of PGE_2_ were concentration-dependent (Figure 1D). On the basis of these results, we examined the effects of co-stimulation with 100 μM ATP and 1 μM PGE_2_ for 5 min in the following experiments.

### 2.2. Involvement of Gi-Coupled EP3R in Synergistic Degranulation Induced by PGE_2_ and ATP

The biological effects of PGE_2_ are known to be mediated by four different EP receptors. Quantitative reverse transcription-polymerase chain reaction (qRT-PCR) revealed that the BMMCs used in this study expressed EP1, EP3, and EP4 receptor mRNAs, while pharmacological experiments with selective EPR antagonists revealed that only the EP3 antagonist ONO-AE3-208 inhibited the degranulation induced by PGE_2_ and ATP (Figure 2A). Consistently, only the ONO-AE-248 agonist against EP3R, a Gi-coupled receptor, induced degranulation in the presence of ATP (Figure 2B). Moreover, the degranulation induced by PGE_2_ and ATP was abolished by pretreating the BMMCs with 50 ng/mL of PTX (Figure 2C).

### 2.3. Involvement of Ionotropic P2X4R in the Effect of ATP and PGE_2_ on Degranulation

We next attempted to identify the P2 receptor subtype that mediates the effect of ATP on degranulation with PGE_2_. We previously reported that under our experimental conditions, BMMCs express ionotropic P2X1, 4, and 7, which are all stimulated by ATP, and G protein-coupled P2Y_1, 2, and 14_ receptors, which are stimulated by ADP, UTP, and UDP-glucose, respectively [21]. Since UDP-glucose had little effect (Figure 1A), we examined the effects of ADP and UTP on degranulation with PGE_2_. As shown in Figure 3A, ADP and UTP had weak effects on degranulation with PGE_2_ compared to ATP. Consistently, the effect of ATP on degranulation with PGE_2_ was not affected by the P2Y_1_ antagonist MRS2179 or P2Y_2_ antagonist AR-C118925 (Figure 3B). Among the P2X receptor antagonists, degranulation induced by ATP and PGE_2_ was inhibited by the P2X4 antagonist 5-BDBD, but not the P2X1 antagonist NF449 or P2X7 antagonist AZ10606120 (Figure 3C,D). Furthermore, the effect of ATP on degranulation with PGE_2_ was enhanced by the P2X4R positive allosteric modulator ivermectin (Figure 3D) but totally absent in BMMCs prepared from P2X4R-deficient mice (Figure 3E).

### 2.4. Mechanism underlying the Synergistic Degranulation Induced by ATP and PGE_2_

We recently reported that stimulating P2X4R enhanced Ag-induced degranulation and increased Src tyrosine kinase signaling pathways, such as Syk and phospholipase Cγ (MS submitted). Therefore, we examined effects of the Src family tyrosine kinase inhibitor PP2 and the Syk inhibitor R406 on degranulation induced by ATP and PGE_2_. As shown in Figure 4A,B, although both PP2 and R406 effectively inhibited the degranulation induced by co-stimulation with ATP and Ag, they had little effect on that induced by ATP and PGE_2_. It has been reported that MC degranulation induced by PGE_2_ is mediated by PI3K [10]. Consistent with the previous report, MC degranulation induced by co-stimulation with ATP and PGE_2_ was inhibited by the non-selective PI3 kinase inhibitor LY294002 but not the structurally similar negative control LY303511 (Figure 4C). The response was also inhibited by AS605240, a specific PI3Kγ inhibitor that is activated by G protein βγ subunits (Figure 4D), whereas the Akt inhibitor triciribine had no effect on the degranulation induced by ATP and PGE_2_ (Figure 4E). These results suggest that P2X4R stimulation enhanced EP3R-mediated signaling in a PI3K-dependent but Akt-independent manner.

### 2.5. Effects of Co-Stimulating BMMCs with ATP and PGE_2_ on ERK1/2, Akt, and Syk Phosphorylation

As shown previously, ATP enhanced Ag-induced Syk phosphorylation (Figure 5A) in BMMCs; however, neither PGE_2_ nor co-stimulation with PGE_2_ and ATP affected Syk phosphorylation (Figure 5A). In contrast, stimulation with PGE_2_ and ATP alone induced ERK1/2 and Akt phosphorylation in a time-dependent manner. Furthermore, Syk phosphorylation in response to ATP and Ag was greater than the response to either alone, whereas the Akt and ERK1/2 phosphorylation induced by co-stimulation with ATP and PGE_2_ was slightly lower (Figure 5B).

### 2.6. Effects of Co-Stimulating BMMCs with ATP and PGE_2_ on [Ca^2+^]i

In Fura-2-loaded BMMCs, ATP stimulation induced a rapid increase in [Ca^2+^]i which decreased to a sustained steady state, whereas PGE_2_ induced a similar [Ca^2+^]i increase, but to a weaker extent than that induced by ATP. Co-stimulation with ATP and PGE_2_ resulted in a sustained increase in [Ca^2+^]i that was greater than the additive responses elicited by ATP or PGE_2_ alone, but was not evident in BMMCs obtained from P2X4-deficient mice (Figure 6A,B), and was inhibited by the PI3Kγ inhibitor AS605240 (Figure 6C,D).

### 2.7. Role of P2X4R Signaling in PGE_2_-Induced Skin MC Activation in Vivo

Administering PGE_2_ to mouse ears reportedly induces edema due to increased extravasation via the EP3R-mediated activation of resident MCs [22]; therefore, we examined whether P2X4R contributed toward this response. Intradermally injecting PGE_2_ into the auricle of wild type mice caused significant Evans blue leakage compared to solvent-injected auricles (Figure 7); however, these effects were absent in MC-deficient Kit^Wsh/Wsh^ mice, confirming that MCs are involved in the PGE_2_-induced effects. Moreover, the effect of PGE_2_ on dye leakage was significantly abrogated in P2X4R-deficient mice (Figure 7).

## 3. Discussion

Previously, we reported that Ag-induced MC degranulation was enhanced by extracellular ATP [12]. Although various GPCR agonists have been shown to enhance Ag-induced degranulation, particularly those acting on Gi-coupled receptors [9,11], we found that the ATP-induced response was mediated by the ionotropic P2X4R [12]. This study examined whether P2X4R stimulation affected IgE-independent MC activation in combination with the GPCR agonists S1P, compound 48/80, UDP-glucose, C5a, histamine, PGD_2_, and PGE_2_, which reportedly activate MCs via the S1P_2_, MRGPRB2, P2Y_14_, C5a, H_3_, DP2, and EP3 receptors, respectively [6]. In this study, these agonists had little effect on MC degranulation alone, with only PGE_2_ causing a strong degranulation response when co-stimulated with ATP. Since ATP is known to promote PG production in various tissues [23,24], this finding may help understand the physiological functions of MCs.

The biological effects of PGE_2_ are mediated by four different receptor subtypes, EP1, EP2, EP3, and EP4Rs [25]. EP3R has been well characterized as the site of action via which PGE_2_ stimulates MC degranulation alone or in the presence of Ag [9,10,22]. In this study, pharmacological investigation using receptor subtype selective agonists and antagonists indicated that the costimulatory effects of PGE_2_ and ATP on degranulation were also mediated by EP3R. In particular, the effect of PGE_2_ was inhibited by the selective EP3 antagonist ONO-AE3-208 and mimicked by the selective EP3 agonist ONO-AE-2248. Furthermore, our observation that the effect of PGE_2_ was abolished by the PTX-mediated inactivation of Gi-dependent signals is consistent with the fact that EP3R is the only Gi-coupled EP receptor subtype [25]. With respect to ATP receptors, we previously demonstrated that BMMCs express mRNA for the P2X1, P2X4, P2X7, P2Y_1_, P2Y_2_ and P2Y_14_ receptors [12]. Several results obtained in this study clearly suggest that the effect of ATP was mediated by P2X4R, similar to enhanced Ag-induced degranulation [12]. First, combining PGE2 with the P2Y_1_ agonist ADP, P2Y_2_ agonist UTP, and P2Y_14_ agonist UDP-glucose did not mimic the effect of ATP on degranulation. In addition, the effect of ATP and PGE_2_ was affected by neither the P2Y_1_ antagonist MRS2179 nor the P2Y_2_ antagonist AR-C118925, indicating that G protein-coupled P2Y receptors are not involved in ATP and PGE_2_-induced degranulation. Moreover, degranulation induced by ATP and PGE_2_ was inhibited by the P2X4 antagonist 5-BDBD but not the P2X1 antagonist NF449 or P2X7 antagonist AZ10606120, and was enhanced by the P2X4R positive allosteric modulator ivermectin [26]. Finally, the degranulation induced by ATP and PGE_2_ was impaired in BMMCs prepared from P2X4R-deficient mice, suggesting that co-activation of Gi-coupled EP3R and ionotropic P2X4R induces MC degranulation.

Previously, we demonstrated that P2X4R stimulation promotes Ag-induced tyrosine phosphorylation signaling, providing a possible mechanism for synergistic MC degranulation in response to Ag and ATP [13]. However, this mechanism was not involved in the effect of ATP and PGE_2_ on MC degranulation, since the response to ATP and PGE_2_ was insensitive to the tyrosine kinase inhibitor PP2 or Syk inhibitor R406, both of which inhibited the synergistic degranulation induced by Ag and ATP. Furthermore, ATP, PGE_2_, and their combination failed to induce Syk phosphorylation. It has been demonstrated that stimulating EP3R with PGE_2_ in BMMCs induces Gβγ-mediated PLCγ and PI3Kγ activation, increased phosphorylation of Akt (a kinase downstream of PI3K), and functionally promotes Ca^2+^ influx [9,11]. In this study, degranulation induced by ATP and PGE_2_ was inhibited by the non-selective PI3K inhibitor LY294002 and PI3Kγ selective inhibitor AS605240; however, ATP did not enhance PGE_2_-induced Akt phosphorylation but rather decreased it slightly. Although ATP itself slightly stimulates Akt phosphorylation, this effect is unchanged in P2X4R KO mice [13], suggesting that ATP-induced Akt phosphorylation is mediated by a receptor other than P2X4R and that P2X4R stimulation does not affect Gβγ-mediated PI3Kγ activation. These results suggest that PI3Kγ activation is necessary for ATP and PGE_2_-induced degranulation but that the downstream mechanism differs to that described for the combination of Ag and PGE_2_ [9,11].

In the Ca^2+^ assay, we observed an enhanced [Ca^2+^]i response to ATP and PGE_2_ that was abrogated in BMMCs obtained from P2X4R KO mice and partly suppressed by inhibiting PI3Kγ with AC605240. Assuming that Ca^2+^ influx through P2X4R is crucial for degranulation in response to ATP and PGE_2_, two potential mechanisms could be inferred. First, since it has been reported that P2X4R channel activity is enhanced by membrane PI(3,4,5)P3 accumulation [27], EP3R-mediated PI3Kγ activation may cause PI(3,4,5)P3 accumulation which in turn promotes P2X4R-mediated Ca^2+^ influx. Second, MCs have been reported to possess functional G protein-coupled inwardly-rectifying K^+^ channel (GIRK), a Gβγ-gated K^+^ channels [28]; therefore, EP3R activation may open these channels, hyperpolarizing the membrane and increasing in driving force of Ca^2+^ influx [29] through the P2X4 receptor. Similar methods of regulation have been demonstrated in Ag-induced MC activation, where Ca^2+^-activated K^+^ channels play a critical role in facilitating Ca^2+^ influx via store-operated Ca^2+^ channels [30]. Further research using electrophysiological analysis is required to explore the precise mechanism of crosstalk between EP3 and P2X4R signaling.

Local PGE_2_ administration has been reported to enhance vascular permeability by directly activating MCs via EP3R activation [22]. In this study, a similar effect of PGE_2_ was reproduced in WT mice but not MC-deficient Kit ^Wsh/Wsh^ mice, while the PGE_2_-induced increase in vascular permeability was shown to be significantly attenuated in P2X4R-deficient mice. These results indicate that ATP positively controls the responsiveness of MCs to PGE_2_ via P2X4R *in vivo*. Since both ATP and PGE_2_ are extracellular mediators that accumulate in damaged or inflamed tissues, the response described here may help to understand the role of MCs in Ag-independent hypersensitivity [14].

Recent accumulating evidence suggest that P2X4R signal is involved in several inflammatory diseases, including neuropathic pain induced by nerve injury [31], joint inflammation in rheumatoid arthritis [32], rejection to transplanted tissue [33], and allergic airway inflammation [34]. The present study focused on the MC degranulation response, an early event in MC activation. As mentioned earlier, MCs can produce wide variety of cytokines [16], which affect chronic responses related to inflammatory diseases. It is therefore important to examine whether P2X4R signaling would affect the MC cytokine production.

In summary, this study demonstrated that co-stimulating MCs with ATP and PGE_2_ synergistically induces degranulation via ionotropic P2X4R and Gi-coupled EP3R, respectively (Figure 8). Moreover, this reaction is independent of the tyrosine kinase cascade, which plays a major role in the Ag-induced response, and is likely to involve enhanced Ca^2+^ influx in a manner dependent on EP3R-mediated PI3K activation. Previously, we reported that ATP also promotes Ag-induced allergic reactions via P2X4R activation; therefore, P2X4R signaling is suggested to act as an enhancer for both Ag-dependent and -independent MC activation. Taken together, targeting the ionotropic P2X4R may be a novel strategy for controlling allergic reactions.

## 4. Materials and Methods

### 4.1. Materials

UTP, ATP, ADP, PGE_1_, PGE_2_, 2,4-dinitrophenyl human serum albumin (DNP-HSA), anti-DNP IgE (clone SPE-7), *p*-nitrophenyl *N*-acetyl-β-d-glucosaminide, and the GenElute Mammalian Total RNA miniprep kit were obtained from Sigma-Aldrich (Tokyo, Japan). Allophycocyanin (APC)-conjugated rat anti-mouse c-Kit antibodies (clone 2B8) were obtained from BD Pharmingen (Tokyo, Japan). Phycoerythrin (PE)-conjugated mouse anti-mouse FcεRIα antibodies (clone MAR-1) were obtained from eBioscience (San Diego, CA, USA). Recombinant mouse interleukin (IL)-3 and recombinant mouse SCF were obtained from Peprotech (London, UK). MRS2179, AR-C118925, NF449, AZ10606120, 5-BDBD, Ivermectin, PP2, LY303511, LY294002, Triciribine, and AS605240 were obtained from Tocris Bioscience (Bristol, UK). R406 was obtained from Cayman Chemical (Michigan, USA). Fura-2-acetoxymethylester (AM) and PTX were obtained from Wako (Osaka, Japan). Anti-phospho-Syk, anti-Syk, anti-phospho-ERK1/2, anti-ERK1/2, anti-phospho-Akt, and anti-Akt antibodies were obtained from Cell Signaling Technology (Danvers, MA, USA). ONO-DI-004 (selective EP1 agonist), ONO-AE1-259-01 (selective EP2 agonist), ONO-AE-248 (selective EP3 agonist), and ONO-AE1-329 (selective EP4 agonist) were obtained from ONO Pharmaceuticals (Osaka, Japan). All other chemicals were of reagent-grade or the highest quality available.

### 4.2. Animals

P2X4R-deficient mice were generated by Dr. Yamamoto (University of Tokyo), as described previously [35]. C57BL/6 and Kit^Wsh/Wsh^ mice were purchased from SLC Japan (Hamamatsu, Japan) and RIKEN (Ibaraki, Japan), respectively. All mice were maintained under specific pathogen-free conditions at the animal facility of Takasaki University of Health and Welfare. All experiments were performed in accordance with the regulations of the Animal Research Committee of Takasaki University of Health and Welfare.(Approval number: 1813, 1 April, 2018)

### 4.3. Cell Culture

BMMCs were established using bone marrow from C57BL/6 mice, as described previously [36]. Briefly, bone marrow cells were collected from the femur and cultured in RPM I1640 medium containing 10% fetal bovine serum, 100 units/mL penicillin, 100 μg/mL streptomycin, and 10 ng/mL recombinant IL-3. After 2 weeks, the cells were cultured with 10 ng/mL of recombinant SCF for 4–6 weeks. After these treatments, almost all (>95%) cells displayed an MC phenotype, as indicated by CD117 (c-Kit) and FcεRI expression measured using a FACSCant II flow cytometer (BD Biosciences, Tokyo, Japan).

### 4.4. Degranulation Assay

Degranulation was evaluated by measuring β-hexosaminidase release, as described previously [36]. BMMCs were sensitized with 50 ng/mL anti-DNP IgE overnight in RPMI 1640 growth medium. Cells were washed twice with phosphate buffered saline (PBS), suspended in Krebs-Ringer-HEPES (KRH) buffer (130 mM NaCl, 4.7 mM KCl, 4.0 mM NaHCO_3_, 1.2 mM KH_2_PO_4_, 1.2 mM MgSO_4_, 1.8 mM CaCl_2_, 11.5 mM glucose) and 10 mM HEPES (pH 7.4) containing 0.1% bovine serum albumin (BSA), and then stimulated under various conditions for 5 min at 37 °C. Reactions were terminated by placing the cells on ice and centrifuging them. Supernatants were collected and the cell pellets were lysed in 1 % Triton X-100. The supernatant and cell lysate were incubated with an equal volume of 1 mM p-nitrophenyl N-acetyl-β-D-glucosaminide dissolved in citrate buffer (pH 4.5) in a 96-well plate at 37 °C for 30 min. Na_2_CO_3_/NaHCO_3_ buffer (pH 10.4) was then added and the absorbance was measured at 405/655 nm.

### 4.5. [Ca^2+^]i Measurement

Cells were collected and washed twice with KRH containing 0.1 % BSA, suspended in KRH-BSA buffer, and treated with 1 μM Fura-2 AM at 37 °C for 20 min. The Fura-2-loaded cells were washed twice with KRH-BSA buffer and adjusted to 1–2 × 10^5^ cells/mL. Changes in Fura-2 fluorescence were measured as described previously [36].

### 4.6. Western Blot

Cells were collected, washed with PBS, and resuspended in KRH. The reaction was performed in KRH buffer and terminated by adding 4× sample buffer. The lysate was separated by 10% sodium dodecyl sulfate polyacrylamide gel electrophoresis (SDS-PAGE) and transferred to Immobilon-P polyvinylidene fluoride (PVDF) membranes. The PVDF membranes were blocked with 5% BSA-TBST for 1 h at room temperature, exposed to primary antibodies overnight at 4 °C, and secondary antibodies for 2 h at room temperature. Antibodies were diluted as follows: anti-phospho-Syk (1:1000), anti-syk (1:1000), anti-phospho-ERK1/2 (1:1000), anti-ERK1/2 (1:1000), anti-phospho-Akt (1:1000), anti-Akt (1:1000), and horseradish peroxidase (HRP)-linked anti-rabbit IgG (1:10000). The entire western blots were shown in Appendix A. 

### 4.7. Quantitative RT-PCR (qRT-PCR)

Total RNA was isolated using the GenElute Mammalian Total RNA miniprep kit. First-strand cDNA was synthesized using Moloney Murine Leukemia Virus reverse transcriptase with a 6-mer random primer. qRT-PCR was performed using an SYBR green kit, as described previously [37].

### 4.8. PGE_2_-Induced Skin Edema

Mice were anesthetized with isoflurane, injected intravenously with 200 μL 0.5% Evans blue diluted in PBS, and injected intradermally in the right ear with PGE_2_ (1.5 nmol) in 20 μL saline and in the left ear with a vehicle of 0.1% ethanol in saline. After 30 min, the mice were sacrificed and their ears collected and weighed. Evans blue dye was extracted from the ears with 1 mL formamide at 55 °C for 24 h and absorbance measured at 620 nm. Data are expressed as μg of Evans blue per mg of ear.

### 4.9. Statistics

All experiments were repeated at least three times, yielding similar results. Data represent the mean ± standard error of the mean (SEM). Statistical analyses were performed using the Student’s *t*-test for two sample comparisons and one-way analysis of variance (ANOVA) with Dunnett’s two-tailed test for multiple comparisons. *p* values < 0.05 were considered statistically significant.

## Figures and Tables

**Figure 1 ijms-20-05157-f001:**
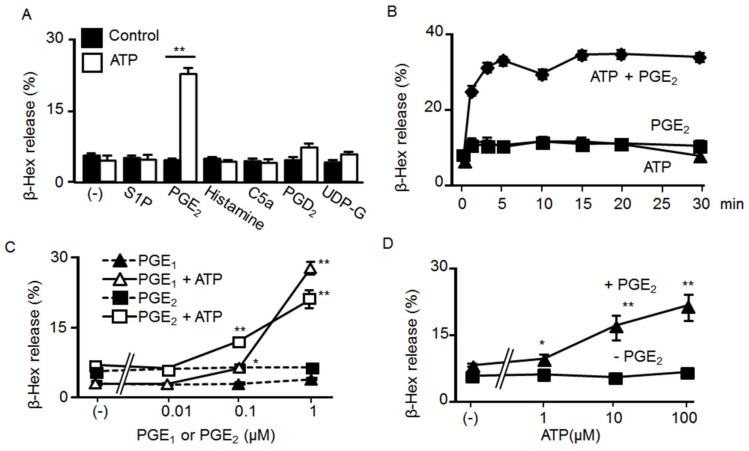
Synergistic effects of ATP and prostaglandin (PG)E_2_ on mast cell (MC) degranulation. (**A**) Bone marrow-derived MCs (BMMCs) were stimulated with sphingosine-1-phosphate (S1P) (1 μM), PGE_2_ (1 μM), histamine (100 μM), C5a (10 nM), PGD_2_ (1 μM), and UDP-glucose (100 μM) with or without ATP (100 μM) (*n* = 3, mean ± SEM). ** *p* < 0.01 indicates a significant difference from the control. (**B**) BMMCs were stimulated for 1–30 min with ATP (100 μM, ▲) and PGE_2_ (1 μM, ■) alone or simultaneously (♦; *n* = 3, mean ± SEM). (**C**) BMMCs were stimulated with different concentrations (0.01–1 μM) of PGE_1_ (△, ▲) and PGE_2_ (□, ■) with (△, □) or without (▲, ■) ATP (100 μM) (*n* = 3, mean ± SEM). * *p* < 0.05 and ** *p* < 0.01 indicate significant differences compared to ATP alone. (**D**) BMMCs were stimulated with different concentrations of ATP (1–100 μM) with (▲) or without (■) PGE_2_ (1 μM) (*n* = 3, mean ± SEM). * *p* < 0.05 and ** *p* < 0.01 indicate significant differences compared to PGE_2_ alone.

**Figure 2 ijms-20-05157-f002:**
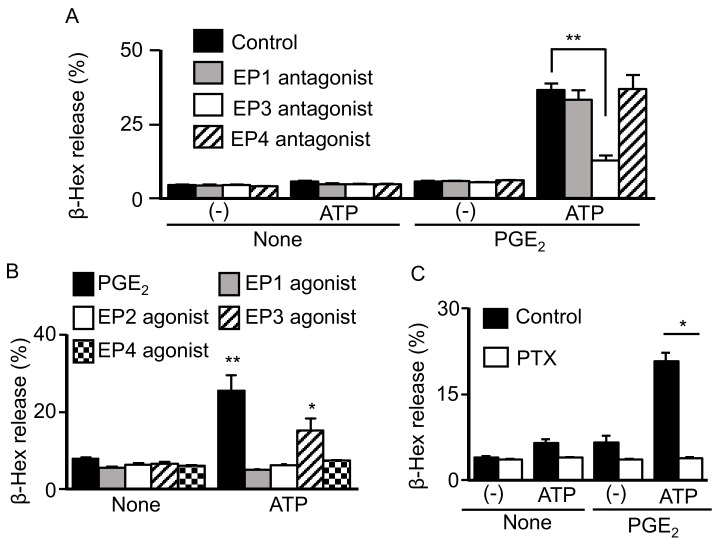
Involvement of EP3 receptor activation in the synergistic effect of prostaglandin (PG)E_2_ and ATP on mast cell (MC) degranulation. (**A**) Bone marrow-derived MCs were preincubated with a vehicle, ONO-8713 (EP1 antagonist), ONO-AE3-240 (EP3 antagonist), and ONO-AE3-208 (EP4 antagonist) at 1 μM for 5 min and then stimulated with vehicle (-) or ATP (100 μM) with or without PGE_2_ (1 μM) for 5 min. Data are shown as the mean ± SEM (*n* = 3). * *p* < 0.05 indicates a significant difference from the control. (**B**) BMMCs were stimulated with PGE_2_, ONO-DI-004 (EP1 agonist), ONO-AE1-259 (EP2 agonist), ONO-AE-248 (EP3 agonist), or ONO-AE1-329 (EP4 agonist) at 1 μM with or without ATP (100 μM). Data are shown as the mean ± SEM (*n* = 3). * *p* < 0.05 and ** *p* < 0.01 indicate significant differences from the response without ATP (none). (**C**) BMMCs were treated with or without pertussis toxin (PTX, 50 ng/mL) overnight and stimulated with ATP (100 μM) with or without PGE_2_ (1 μM) for 5 min. Data are shown as the mean ± SEM (*n* = 3). * *p* < 0.05 indicates a significant difference from the control.

**Figure 3 ijms-20-05157-f003:**
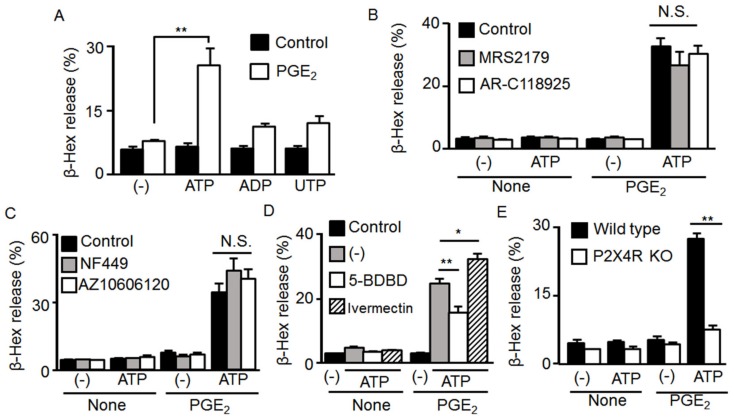
Involvement of P2X4 receptor (P2X4R) activation in the synergistic effect of prostaglandin (PG) E_2_ and ATP on mast cell (MC) degranulation. (**A**) Bone marrow-derived MCs (BMMCs) were stimulated concurrently with PGE_2_ (1 μM) and a vehicle, ATP, ADP, or UTP (100 μM) for 5 min. (**B**) BMMCs were preincubated with a vehicle, the P2Y_1_ antagonist MRS2179 (10 μM), or the P2Y_2_ antagonist AR-C118925 (10 μM) for 5 min and then stimulated with ATP (100 μM) with or without PGE_2_ (1 μM) for 5 min. (**C**) BMMCs were preincubated with a vehicle, the P2X1 antagonist NF449 (10 μM), or the P2X7 antagonist AZ10606120 (1 μM) for 3 min and then stimulated with ATP (100 μM) with or without PGE_2_ (1 μM) for 5 min. (**D**) BMMCs were preincubated with a vehicle, the P2X4 antagonist 5-BDBD (10 μM), or the P2X4R positive allosteric modulator ivermectin (10 μg/mL) for 5 min and then stimulated with ATP (100 μM) with or without PGE_2_ (1 μM) for 5 min. (**E**) BMMCs prepared from wild type and P2X4R-deficient mice (P2X4R KO) were stimulated with ATP (100 μM) with or without PGE_2_ (1 μM) for 5 min. Data are shown as the mean ± SEM (*n* = 3). N.S. no significant difference, * *p* < 0.05 and ** *p* < 0.01 indicate significant differences.

**Figure 4 ijms-20-05157-f004:**
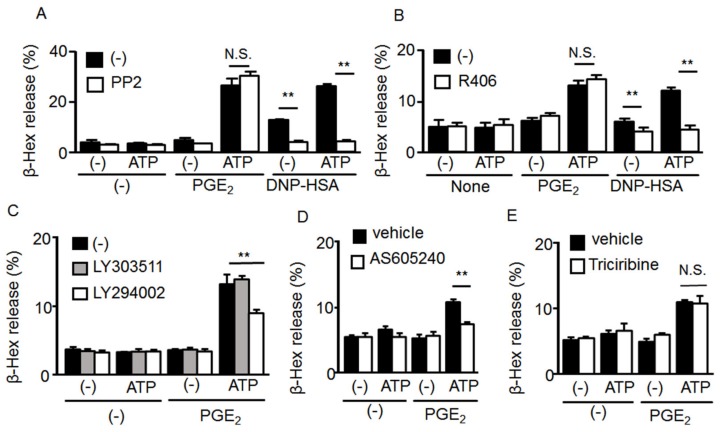
Effect of tyrosine kinase and phosphoinositide 3-kinase (PI3K)/Akt signaling pathway inhibitors on mast cell (MC) degranulation induced by co-stimulation with ATP and prostaglandin (PG)E_2_. (**A**) Bone marrow-derived MCs (BMMCs) were preincubated with a vehicle or the Src tyrosine kinase inhibitor PP2 (1 μM) for 5 min and then stimulated with ATP (100 μM) with or without PGE_2_ (1 μM) or 2,4-dinitrophenyl human serum albumin (DNP-HSA, 10 ng/mL). (**B**) BMMCs were preincubated with a vehicle or the Syk inhibitor R406 (2 μM) for 5 min and then stimulated with ATP (100 μM) with or without PGE_2_ (1 μM) (*n* = 3). (**C**) BMMCs were preincubated with a vehicle, the PI3K inhibitor LY294002 (10 μM), or the control compound LY303511 (10 μM) for 5 min and then stimulated with ATP (0.1 mM) with or without PGE_2_ (1 μM) (*n* = 3). (**D**) BMMCs were preincubated with a vehicle or the PI3Kγ inhibitor AS605240 (1 μM) for 5 min and then stimulated with ATP (100 μM) with or without PGE_2_ (1 μM) (*n* = 3). (**E**) BMMCs were preincubated with a vehicle or the Akt inhibitor triciribin (10 μM) for 5 min, and then stimulated with ATP (100 μM) with or without PGE_2_ (1 μM) (*n* = 3). Data are shown as the mean ± SEM. N.S. no significant difference, ** *p* < 0.01 indicates a significant difference.

**Figure 5 ijms-20-05157-f005:**
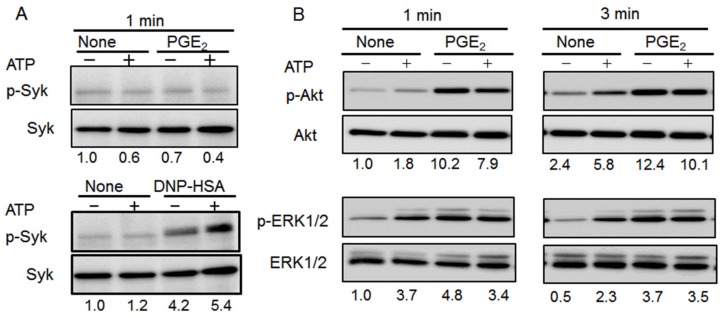
Effect of co-stimulation with ATP and prostaglandin (PG)E_2_ on Syk, extracellular signal-regulated kinase (ERK)1/2, and Akt phosphorylation in bone marrow-derived mast cells (BMMCs). (**A**) BMMCs were stimulated with ATP (100 μM) with or without PGE_2_ (1 μM, upper) or 2,4-dinitrophenyl human serum albumin (DNP-HAS,10 ng/mL, lower) for 1 min. Cell lysates were subjected to western blot analysis for phospho-Syk and total-Syk. (**B**) BMMCs were stimulated with ATP (100 μM) with or without PGE_2_ (1 μM) for 1 (left) or 3 (right) min. Cell lysates were subjected to western blot analysis for phospho-Akt and total Akt (upper) or phospho-ERK 1/2 and total-ERK 1/2 (lower). The numbers below each image indicate normalized relative phosphorylated protein intensity; the results for no stimulation are set to one. Blots are representative of three independent experiments.

**Figure 6 ijms-20-05157-f006:**
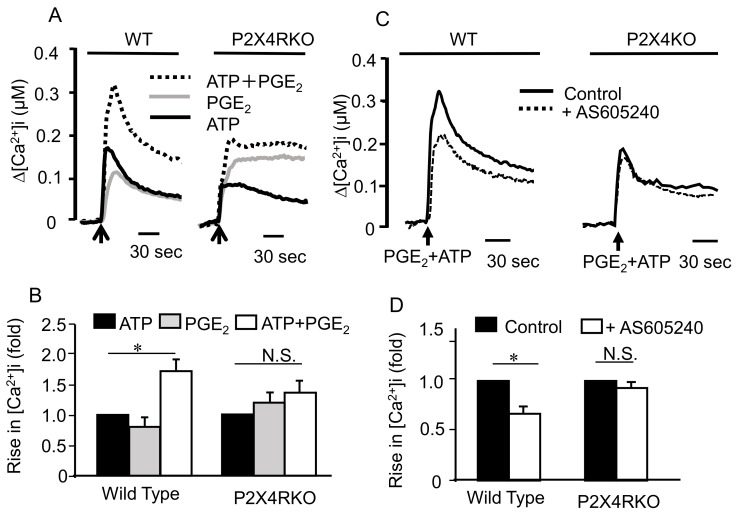
Effects of co-stimulating bone marrow-derived mast cells (BMMCs) with ATP and prostaglandin (PG)E_2_ on intracellular Ca^2+^ concentration ([Ca^2+^]i) levels. (**A**) BMMCs prepared from widg type (WT) or P2X4 receptor deficient (P2X4RKO) mice were loaded with Fura-2 acetoxymethyl ester and changes in [Ca^2+^]i were monitored after stimulating with ATP (black line), PGE_2_ (gray line), or ATP plus PGE_2_ (dotted line) at the time indicated by the arrow. The Ca^2+^ data are representative of four independent experiments. (**B**)Summary of the data obtained in A. Results are indicated as fold of ATP-induced response (WT; 205 ± 38 nM, *n* = 4, P2X4RKO; 115 ± 16 nM, *n* = 4). Data are shown as the mean ± SEM (*n* = 4). N.S.; no significant difference, * *p* < 0.05 indicates a significant difference. (**C**) BMMCs prepared from WT or P2X4RKO were preincubated with or without the PI3Kγ inhibitor AS605240 (1 μM) for 5 min and then stimulated with ATP (100 μM) plus PGE_2_ (1 μM). The superimposed [Ca^2+^]i changes are representative of at least four different BMMC preparations obtained from different animals. (**D**) Summary of the data obtained in C. Results are indicated as fold of control response (WT; 342 ± 45 nM, *n* = 3, P2X4RKO; 158 ± 29 nM, *n* = 4). Data are shown as the mean ± SEM (*n* = 3–4). N.S.; no significant difference, * *p* < 0.05 indicates a significant difference.

**Figure 7 ijms-20-05157-f007:**
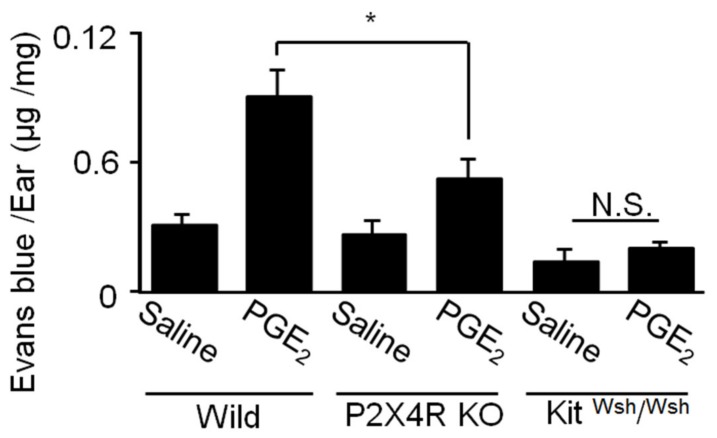
Comparison of prostaglandin (PG) E_2_-induced hyperpermeability in wild type, P2X4 receptor-deficient (P2X4RKO), and mast cell-deficient Kit ^Wsh/Wsh^ mice. PGE_2_ (1.5 nmol) was intradermally injected into the ear and vascular permeability measured 30 min later. Data are shown as the mean ± SEM (*n* = 5–6). N.S. no significant difference, * *p* < 0.05 indicates a significant difference.

**Figure 8 ijms-20-05157-f008:**
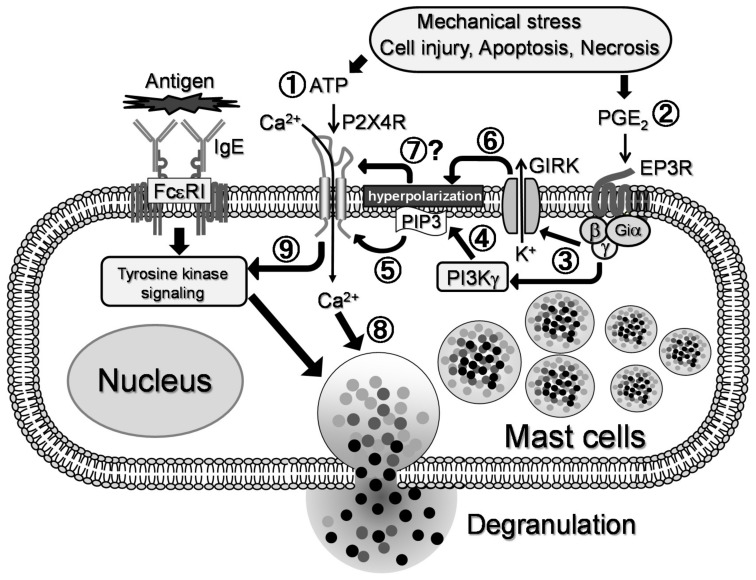
Proposed mechanism of interaction between P2X4 receptor (P2X4R) and EP3 receptor (EP3R) signals for the synergistic degranulation in mast cells(MCs). Extracellular ATP released from damaged cells stimulates MC P2X4R ①, leading to Ca^2+^ influx. Such conditions are accompanied by an inflammation with increased production of prostaglandin (PG)E_2_. In MCs, PGE_2_ stimulates Gi-coupled EP3R ②, leading to activation of phosphoinositide 3-kinase (PI3K)γ and G protein-coupled inwardly-rectifying K^+^ channel (GIRK) via βγ-complex of the G protein ③. Activation of PI3Kγincreases phosphoinositide-3,4,5-trisphosphate (PIP3) levels in plasma membrane ④, thereby promoting P2X4R channel activity ⑤. GIRK activation may cause hyperpolarization of the membrane potential ⑥, which would increase the driving force of Ca^2+^ inflow through P2X4R ⑦. These interactions between P2X4R and EP3R signals lead to the observed synergy in MC degranulation ⑧. P2X4R signal also promotes IgE-dependent tyrosine kinase-mediated signals to induce facilitated degranulation, as described previously ⑨ [12].

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
