# Peer review of "Co-Stimulation of Purinergic P2X4 and Prostanoid EP3 Receptors Triggers Synergistic Degranulation in Murine Mast Cells"

_ijms, 2019, doi:10.3390/ijms20205157_

Round 1
Reviewer 1 Report
- Title: Co-stimulation of purinergic P2X4 and prostanoid 3 EP3 receptors triggers synergistic degranulation in 4 mouse bone marrow-derived mast cells.
- In this paper the authors study the degranulation of mast cells and its activation receptors in hypersensitivity in mice. The authors conclude that P2X4R signaling improves the EP3R-mediated MC activation through a mechanism different from that involved in the enhancement responses induced by Ag.
- The title is a little too long.
- The article seems to me complete and worthy of publication, however, there are some corrections to be made. For example, I would insert a diagram of the biochemical cascade of activation of mast cells. When the antigen binds to the FceRI receptor what happens before producing pro-inflammatory cytokines and prostaglandins? This scheme is necessary for a greater understanding of the reader.
- This article must be usable not only to the experts of mast cells, which are not very many in the world, but also to the authors of other fields of medicine. For this reason I ask to simplify some general information on mast cells and their production of inflammatory cytokines.
- In the light of above observations the authors should briefly describe this theme. Therefore, to make this paper more complete and interesting for the readers of this important journal, the authors should expand a bit the introduction and discussion on this subject. Here, three important articles have recently been reported. Below I list this interesting articles that should be studied, incorporate the meaning and report briefly in the paper and in the list of references.
- The mast cell - neurofibromatosis connection. Antonopoulos D, Tsilioni I, Balatsos NAA, Gourgoulianis KI, Theoharides TC. J Biol Regul Homeost Agents. 2019 May-Jun,;33(3):657-659.
- Impact of mast cells in systemic lupus erythematosus: can inflammation be inhibited? Caraffa Al, Gallenga CE, Kritas SK, et al. J Biol Regul Homeost Agents. 2019 May-Jun;33(3):669-673.
- Interleukin-1 family cytokines and mast cells: activation and inhibition. Gallenga CE, Pandolfi F, Caraffa Al, et al. J Biol Regul Homeost Agents. 2019 Jan-Feb,;33(1):1-6.
- I believe these suggestions are important for improving this paper. Without these corrections the paper cannot be published. So I recommend minor revision.
I'd like to review this article after corrections.
Author Response
Thank you for your valuable comments for our manuscript. According to your suggestion, we revised the title, and rewrote introduction with references that you recommended (page 2, lines 68-82). We added Figure 8 (page 10) to explain our results, and added some discussion (page 9, lines 307-313).
Reviewer 2 Report
The authors have conducted and described a thoughtful, elegant study. Experiments are well designed and the readers walk away with a story of mast cell activation by ATP via a novel signaling pathway that modulates both antigen-dependent and antigen-independent mast cell activity. What is absent from the paper is the biological significance of this and why the reader should care! Why is ATP an activating signal? Why does ATP recognition and resulting mast cell activation bypass some of the other modes of G-protein driven mast cell activity? The authors could contextualize this interesting aspect of mast cell physiology using ideas of the danger signal (Polly Matzinger), sentinel functions of mast cells in response to things like tissue damage, necrosis, pyroptosis etc. which might release ATP into the extracellular environment. Mast cells are found within epithelial tissues where ATP release is used as a mechanism for cell volume control (Fitz, 2007) to maintain cell size where tissue integrity depends on epithelial cell tight junctions. What might be the significance of the signaling pathway the authors have described in such a homeostatic scenario? While in vivo (knockout, inducible knockout) experiments to understand the physiological implications of this signaling pathway are likely beyond the scope of this study, the introduction and the discussion should be rewritten to clearly describe its biological significance to the readers thus alerting them to new functions of mast cells in the body.
Author Response
Thank you for your valuable comments for our manuscript. According to your suggestions, we rewrote introduction with reference reported by Fitz (page 2, lines 64-65, and lines 68-82), and added some statements in discussion (page 9, lines 307-313). In addition, Figure 8 is added to give an overview of our results.